# TaxoKnow: Taxonomy as Prior Knowledge in the Loss Function of Multi-class Classification

**Mohsen Pourvali, Yao Meng, Chen Sheng, Yangzhou Du**

Lenovo Research, Beijing, China
{mpourvali, mengyao1, shengchen1, duyz1}@lenovo.com

## Abstract

In this paper, we investigate the effectiveness of integrating a hierarchical taxonomy of labels as prior knowledge into the learning algorithm of a flat classifier. We introduce two methods to integrate the hierarchical taxonomy as an explicit regularizer into the loss function of learning algorithms. By reasoning on a hierarchical taxonomy, a neural network alleviates its output distributions over the classes, allowing conditioning on upper concepts for a minority class. We limit ourselves to the flat classification task and provide our experimental results on two industrial in-house datasets and two public benchmarks, RCV1 and Amazon product reviews. Our obtained results show the significant effect of a taxonomy in increasing the performance of a learner in semi-supervised multi-class classification and the considerable results obtained in a fully supervised fashion.

## Introduction

Large Language Models (LLMs), e.g., BERT and GPT-3, have made significant advances in Natural Language Processing (NLP). In general, pre-training, where a model first trains on massive amounts of data before being fine-tuned for a specific task, has proven to be an efficient technique for improving the performance of a wide range of language tasks (Min et al. 2021).

If we break down the architecture of LLMs, we can categorize their components into two general concepts: Deep Neural Network (DNN) as a part of Machine Learning (ML), and Training Data. Despite all the advantages of LLMs, they come with some limitations. Starting from the very beginning, machine learning has its own limitations, from supervised ML which heavily relies on large amounts of human-labeled data to Reinforcement Learning (RL) which requires a very large number of interactions between the agent and the environment. The brittleness of deep learning systems is largely due to machine learning models being based on the independent and identically distributed (i.i.d.) assumption, which is not a realistic assumption in the real world. In general, Multi-Layer Perceptrons (MLPs) are good at generalizing within the space of training examples, but they perform poorly at generalizing outside the space of training examples, and this limitation is not improved

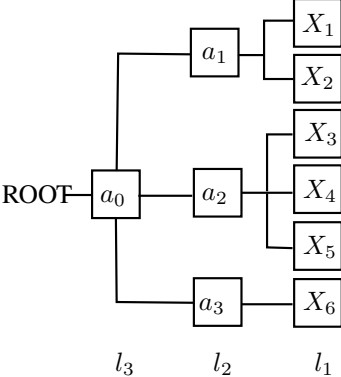

$$\delta_{22} = (\neg X_1 \wedge \neg X_2 \wedge X_3 \wedge \neg X_4 \wedge \neg X_5 \wedge \neg X_6) \vee$$
$$(\neg X_1 \wedge \neg X_2 \wedge \neg X_3 \wedge X_4 \wedge \neg X_5 \wedge \neg X_6) \vee$$
$$(\neg X_1 \wedge \neg X_2 \wedge \neg X_3 \wedge \neg X_4 \wedge X_5 \wedge \neg X_6)$$

Figure 1: A symbolic representation/sentence for node $a_2$ in a higher level $l_2$ of a hierarchical taxonomy for multi-class classification

even by adding more layers. So, the question is, what can be done? Can increasing the size of training data solve these shortcomings?

Another shortcoming, which is not addressed by simply using more data, is *curve fitting* (Pearl 2019), mapping inputs to outputs. If our systems rely solely on curve-fitting and statistical approximation, their inferences will necessarily be shallow (Bender and Koller 2020). Instead of inducing a more abstract and causal understanding of the world, they try to approximate the statistical curves of how words are used to infer how the world works.

Let us take a step back and explore another approach to training a machine, which is *Symbolic Machine* learning. A symbolic machine combines a sophisticated reasoner with a large-scale knowledge base. Knowledge can be formulated in a logical function with symbolic variables, for example, $\delta_{22}$ in Figure 1 expresses that for the set of variables $< X_1, X_2, .., X_6 >$, one and exactly one of $X_3$, $X_4$, or $X_5$ must be true, with the rest being false. One well-known ex-

ample of a symbolic machine is CYC [1], It was launched in 1984 by Doug Lenat and required thousands of person-years of effort to capture facts about psychology, politics, economics, biology, and various other domains in a precise logical form. One famous test of CYC is the Romeo and Juliet quiz, in which CYC demonstrates an internal distillation of a complex scenario and provides an example of rich cognition. However, despite the extensive efforts put into CYC, it falls short compared to the remarkable results achieved by transformers and GPT-2, even without explicit knowledge engineering.

What Gary Marcus (Marcus 2020) believes is that symbol manipulation could be the solution, particularly for extrapolating beyond a training regime. Symbol manipulation, specifically the machinery of operations over variables, offers a natural albeit incomplete solution to the challenge of extrapolating beyond a training regime. It also provides a clear basis for representing structured representations (such as the tree structures foundational to generative linguistics) and records of individuals and their properties. It can bring a hybrid approach that combines the best of both worlds: the ability to learn from large-scale datasets and the capacity to represent abstract concepts. The power of combining statistical and symbolic artificial intelligence techniques to accelerate learning and improve transparency is exemplified by (Mao et al. 2019).

In this work, we aim to integrate abstract/prior knowledge (Hierarchical Taxonomy of labels) into the structure of machine learning. As one of our contributions, we leverage symbolic manipulation to represent the taxonomy. According to Henry Kautz's proposal on Neural-Symbolic Computing (NSC) (Garcez and Lamb 2023), our work can be categorized as type 5; $NOURO_{SYMBOLIC}$; a tightly-coupled neural-symbolic system where a symbolic logic rule is mapped onto a distributed representation (an embedding) and acts as a soft-constraint (a regularizer) on the network's loss function. Additionally, we combine type 5 with a method from type 1; *SYMBOLIC NEURO SYMBOLIC*; which involves standard deep learning in which input and output of a neural network can be made of symbols. Our target is an imbalanced classification problem where we have a Hierarchical Taxonomy of labels as our prior knowledge.

Many real-world classification problems exhibit imbalanced class distributions. In current fully supervised classification tasks, models are trained on labeled datasets where labels are primarily injected into the objective function (e.g., cross-entropy) as prior knowledge. These labels typically originate from a larger hierarchical taxonomy, allowing for comprehensive reasoning over the labels. Labels in Machine Learning (ML), especially in supervised ML, play an important role. However, labels often present challenges. Despite the human cost required for labeling, labels are frequently incomplete, ambiguous, and redundant. Using a hierarchical taxonomy for labels can provide more information that leads to improved labels and ultimately enhances model quality in supervised learning, and even yields further gains in semi-supervised learning.

In this paper, we introduce two methods to represent and incorporate the hierarchical taxonomy. The first method (Section ) represents the taxonomy as constraints in Boolean logic. For example, Figure 1 illustrates a hierarchical taxonomy for class labels, where leaves at level $l_1$ indicate the actual class labels used in the loss function (e.g., cross-entropy), and nodes at a higher level $l_2$ indicate a higher level of conceptualization for the labels, which are typically not used in the classification algorithm. The second method (Section ) involves using Graph Convolutional Networks (GNN) to represent and incorporate the hierarchical taxonomy into the loss function. Our experimental results for both methods demonstrate the significant effect of higher levels of the hierarchical taxonomy in alleviating the unequal distribution of classes in severely imbalanced classification problems[2].

Our contributions in this paper focus on flat/general classification, referring to the standard multi-class classification problem. This differs from hierarchical classification, where the class set to be predicted is organized into a class hierarchy, typically represented as a tree or a Directed Acyclic Graph (DAG).

## Related Work

**Imbalanced Classification:**    Approaches for dealing with imbalanced classification problems can be categorized into three groups: data-level approaches, algorithm-level techniques, and hybrid methods (Johnson and Khoshgoftaar 2019). Data-level approaches aim to address the unequal distribution of classes by employing sampling techniques such as over-sampling the minority class or under-sampling the majority class. However, under-sampling may result in the loss of important information for the model to learn from, while over-sampling can increase training time and lead to overfitting (Johnson and Khoshgoftaar 2019). Algorithm-level techniques, on the other hand, adjust the learning or decision process to give more importance to the minority class. Hybrid methods combine data-level and algorithm-level approaches in various ways to tackle the class imbalance problem (Seiffert et al. 2009; Chen et al. 2021).

**Taxonomy-aware Classification:**    The use of hierarchical concepts in classification has been explored in various studies. For example, (Brust and Denzler 2020) leverages a publicly available hierarchy like WordNet to integrate additional domain knowledge into classification.

Existing works on integrating taxonomy into machine learning can generally be grouped into two approaches. The first approach involves *indirectly* incorporating taxonomy information into the ML model, such as label expansion (Li et al. 2017). The second approach focuses on *directly* integrating taxonomy information into the model architecture (Karamanolakis, Ma, and Dong 2020; Jenkins, Bloom, and Zhang 2021; Ong et al. 2022). (Ong et al. 2022) demonstrates that using taxonomy information of plant species can alleviate class sparsity issues when optimizing for a large

---

[1]www.cyc.com

[2]The code and datasets will be hosted on https://github.com/mpourvali/TaxoKnow.

number of classes. In the domain of semi-supervised learning, (Su and Maji 2021) proposes techniques for incorporating coarse taxonomic labels to train image classifiers in fine-grained domains.

While previous works have explored the effectiveness of label taxonomies in hierarchical classification, our paper emphasizes the positive impact of hierarchical taxonomies in flat classification problems. To the best of our knowledge, our work is the first to propose injecting hierarchical taxonomies of labels as prior knowledge into flat classification problems. Our approach falls under algorithm-level techniques for addressing imbalanced classification problems, as we directly inject a hierarchical taxonomy of class labels as prior knowledge into the existing loss function (i.e., data-driven) of a deep neural network.

## Proposed Methods

We propose two approaches to represent and integrate the hierarchical taxonomy as prior knowledge into the loss function of a learning algorithm.

### Symbolic-based Approach

To integrate the hierarchical taxonomy of the classes into the loss function, we first represent the taxonomy as symbolic logical constraints. Building on the work of (Xu et al. 2018) we derive a differentiable semantic loss function that captures how well the neural network satisfies the constraints on its output.

**General Notation.** We employ concepts in propositional logic to formally define taxonomy and semantic loss. Boolean variables are written in uppercase letters $(X, Y)$, and their instantiation $(X = 0$ or $X = 1)$ are written in lowercase $(x, y)$. We write sets of variables in bold uppercase $(\mathbf{X}, \mathbf{Y})$, and their joint instantiation in bold lowercase $(\mathbf{x}, \mathbf{y})$. A literal is a variable $(X)$ or its negation $(\neg X)$. A logical sentence $(\alpha$ or $\beta)$ is created by variables and logical connectives $(\wedge, \vee, \text{etc.})$, and is also called a formula or constraint. A state $\mathbf{x}$ satisfies a sentence $\alpha$, denoted as $\mathbf{x} \models \alpha$, if the sentence evaluates to be true in that world, as defined in the usual way. The output vector of a neural network is denoted by $\mathsf{p}$, where each value in $\mathsf{p}$ represents a probability of an output in $[0, 1]$. The output vector of a set of sentences is denoted by $\mathsf{s}$, where each value in $\mathsf{s}$ represents a satisfaction value in $[0, 1]$.

**Taxonomy.** Each level of concepts in a taxonomy is denoted as $l_i, i \in [1, K]$, where $K$ is node-based length of the taxonomy and $l_1$ indicates the leaves of the taxonomy, which is associated with the class labels. Each node in taxonomy except nodes in the leaves is denoted as $a_i$. For instance, in Figure 1, for a taxonomy used in multi-class classification, sentence $\delta_{22}$ states that for a set of indicators $\mathbf{X} = \{X_1, .., X_6\}$, one and exactly one of $X_3, X_4, X_5$ must be true, while the rest must be false. This statement indeed represents node $a_2$ of the taxonomy in terms of its children/variables $(X_3, X_4, X_5)$. To represent hierarchical nature of the taxonomy, a set of variables $\mathbf{B} = \{B_1, B_2, .., B_{K-1}\}$ is defined over the taxonomy levels. $B_1, ..., B_{K-1}$ correspond

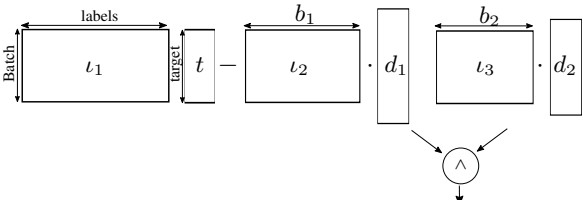

$$CrossEntropy(\iota_1, t) - w \times \log(wmc(\iota_2 \cdot wmc(d_1) \wedge \iota_3 \cdot wmc(d_2)))$$

Figure 2: An illustration for supervised semantic loss. $\iota_1$ is the matrix of model output for a batch, $\iota_2$ is the matrix of one-hot vectors over nodes in level $l_2$ in the batch, and $\iota_3$ is the matrix of one-hot vectors over nodes in level $l_3$ in the batch. $|b_i| = |d_i|$ since each element in one-hot vector $b$ is corresponding to a sentence $\delta$ in $d$.

to the variables of each non-leaf node in the taxonomy tree, where each variable of $\mathbf{B}$ corresponds to a set of one-hot vectors $b_j$, e.g., $b_1$ and $b_2$ correspond to level 1 and level 2, as shown in Figure 2 and Figure 3. $b_j$ corresponds to one-hot vector over $a_0, a_1, ..., a_m$, where $m$ is number of nodes in level $l_j$, e.g., as it is shown in Figure 1 $a_1, a_2, a_3$ for level $l_2$. A logical sentence $\beta$ is created from variables $\mathbf{B}$ and logical connective $\wedge$. For a given taxonomy there would be a sentence $\delta_{ij}$ corresponding to the node $a_i$ (i.e., propositional logic) which all sentences for each level $l_j$ are stored in $d_j$ ($d_1$ and $d_2$ as it's shown in Figure 2), and sentence $\alpha$ is defined over $d_1, d_2, ..., d_{K-1}$.

**Semantic Loss.** The semantic loss $L^s(\alpha, \beta, \mathsf{p}, \mathsf{s})$ is defined as a function of sentences $(\alpha, \beta)$ in propositional logic, which is defined over variables $\mathbf{X} = \{X_1, X_2, .., X_n\}$ and $\mathbf{B} = \{B_1, B_2, .., B_{K-1}\}$, a vector of probabilities $\mathsf{p}$ for variables $\mathbf{X}$, and a satisfaction vector $\mathsf{s}$ for variables $\mathbf{B} = \{B_1, B_2, .., B_{K-1}\}$. The element $\mathsf{p}_i$ denotes the predicted probability of variable $X_i$, corresponding to a single output of the neural network. The element $\mathsf{s}_i$ represents the satisfaction score of variable $B_i$, corresponding to the output of a sentence $\alpha$. Similar to (Xu et al. 2018), we provide two examples of integrating semantic loss $L^s$ into an existing loss function as an additional regularization term, in both supervised and semi-supervised manners. Specifically, with a weight $w$, Equation 1 shows the new loss.

$$existing\_loss + w \cdot semantic\_loss \qquad (1)$$

**Supervised-based Definition.** In the Supervised-based definition, we assume that all the training dataset is labeled, and the hierarchical taxonomy is complete, meaning that for labeled class, all the upper parents are known. Formally, for a class label $cl_i$, its $K - 1$ upper concepts in the taxonomy are given. With this assumption, let $\mathsf{p}$ be a vector of probabilities, one for each variable in $\mathbf{X}$, let $\alpha$ be a sentence over $\mathbf{X}$, and $\beta$ be a sentence over $\mathbf{B}$. Equation 2 represents the hierarchical taxonomy as a logical constraint.

$$L^s(\alpha, \beta, \mathsf{p}, \mathsf{s}) \propto -\log \prod_{\mathbf{y} \models \beta} \sum_{\mathbf{X} \models \alpha} \prod_{i:\mathbf{X} \models X_i} \mathsf{p}_i \prod_{i:\mathbf{X} \models \neg X_i} (1 - \mathsf{p}_i) \quad (2)$$

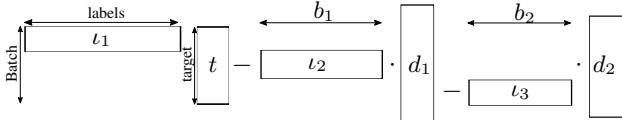

$$CrossEntropy(\iota_1, t) - w_1 \times \log(\iota_2 \cdot wmc(d_1)) - w_2 \times \log(\iota_3 \cdot wmc(d_2))$$

Figure 3: An illustration for semi-supervised semantic loss. $\iota_1$ is the matrix of model output for the labeled (i.e., leaves in taxonomy) data, $\iota_2$ is the matrix of one-hot vectors over nodes in level $l_2$ without labels in leaves, and $\iota_3$ is the matrix of one-hot vectors over nodes in level $l_2$ without labels in leaves and level $l_2$.

where **y** is a state that satisfies $\beta$. By applying the negative logarithm, we enforce the training model to satisfy the constraint. Figure 2 provides an illustration of Equation 2.

Our goal is to develop a tractable loss for computing both semantic loss and its gradient. From propositional logic theories, we know that a Model is a solution to a given propositional formula $\Delta$, and Model Counting or #SAT is the problem of computing the number of models for $\Delta$. In case of mapping literals of the variables to non-negative real-valued weights, we will have Weighted Model Counting (WMC) (Chavira and Darwiche 2008; Sang, Beame, and Kautz 2005). The well-known task of model counting corresponds to the special case where all literal weights are 1 (and counts thus restricted to the natural numbers), whereas probabilistic inference (Prob) in a setting where all variables are independently assigned truth values at random restricts the weight function $\omega$ of WMC to values from [0, 1] such that weights of positive and negative literals for each var sum to one, i.e., for every variable $v$, $\omega(v) \in [0, 1]$ and $\omega(\neg v) = 1 - \omega(v)$ (Kimmig, Van den Broeck, and De Raedt 2017).

From (Darwiche 2003), we know about differential circuit languages that compute WMCs, which are amenable to backpropagation. Following (Xu et al. 2018), we use the circuit compilation techniques from (Darwiche 2011), namely the Sentential Decision Diagram (SDD), to construct a Boolean circuit representing semantic loss. The SDD circuit form exhibits two main properties: determinism and decomposability, allowing us to compute both the values and gradients of the semantic loss in time linear to the size of the circuit (Darwiche and Marquis 2002).

**Semi-supervised-based Definition.** In this section, we demonstrate the integration of a hierarchical taxonomy with unlabeled data. In the Semi-supervised-based definition, the assumption is that there is unlabeled data and the hierarchical taxonomy is not complete. The semantic loss is defined for unlabeled data using an incomplete taxonomy. The labeled data is directly used in an existing loss function (e.g., cross entropy). For the unlabeled data, we employ the available deepest concepts/nodes from the root, and the upper node is considered in case of a missing lower node. In this definition of the semantic loss, since there are no conflicts

between different levels of concepts in the hierarchical taxonomy, there is no need for a sentence $\beta$ over **B**. The intuition behind this is to emphasize the information carried by unlabeled data and provide a level-based weighting for the incomplete taxonomy.

$$L^s(\alpha, \beta, \mathsf{p}, \mathsf{s}) \propto - \log \sum_{j \in \{1, K\}} \sum_{\mathbf{X} \models \alpha} \prod_{i: \mathbf{X} \models X_i} \mathsf{p}_i \prod_{i: \mathbf{X} \models \neg X_i} (1 - \mathsf{p}_i)$$

(3)

Equation 3 is illustrated in Figure 3, including the training batch and SDDs.

In essence, Equation 2 and 3 expand semantic loss (Xu et al. 2018) over hierarchical structure. They are proportional to the negative logarithm of the probability of generating a state that satisfies the constraint when sampling values according to $\mathsf{p}$.

## GCN-based Approach

Graph Convolutional Networks is a powerful method presented for semi-supervised learning on graph-structured data (Kipf and Welling 2016), in which the authors introduced GCN to address the problem of classifying nodes, such as documents, in a graph, such as a citation network, where labels are only available for a small subset of nodes. Similarly, in representing hierarchical taxonomy in semi-supervised learning, we deal with the labeling concept in different levels of the hierarchy. Our objective is to identify representations for some nodes in the taxonomy, given the labels of other nodes. Moreover, the ability of GCN to handle symbolic inputs/outputs offers a differentiable alternative for semantic loss and logical constraints. These two reasons led us to utilize a graph neural network (GCN) for knowledge integration. We consider a hierarchical taxonomy as a labeled graph and seek the GCN encoding of any externally connected node to this graph. Figure 4 illustrates the 2-Dimensional GCN encoding of the nodes in the sample taxonomy.

One issue with GCN is the large memory requirement when encoding a big graph-structured data to provide representations for each node. Moreover, using GCN on the entire graph data avoids the need for explicit regularization with another supervised loss function, such as Cross Entropy. In this section, we propose a method to incorporate the hierarchical taxonomy of a classification task as prior knowledge into the loss function through a Batch-based Graph Convolutional Networks (BGCN). A representation for a graph $A$ in GCN is defined as:

$$H^{(l+1)} = \sigma(\tilde{D}^{-\frac{1}{2}} \tilde{A} \tilde{D}^{-\frac{1}{2}} H^{(l)} W^{(l)})$$

(4)

where $\tilde{A} = A + L_N$ is the adjacency matrix of the undirected graph $A$ with added self-connections. $I_N$ is the identity matrix, $\tilde{D}_{ii} = \sum_j \tilde{A}_{ij}$ and $W^{(l)}$ is a layer-specific trainable weight matrix. $\sigma(.)$ Denotes an activation function (we used $ReLU$ in our experiments). $H^{(l)} \in R$ is the matrix of activations in the $l^{th}$ layer; $H^0 = X$, $H^2 = softmax(H^1)$.

We provide a taxonomy backbone graph for each batch, which is consistent across all batches and is generated from

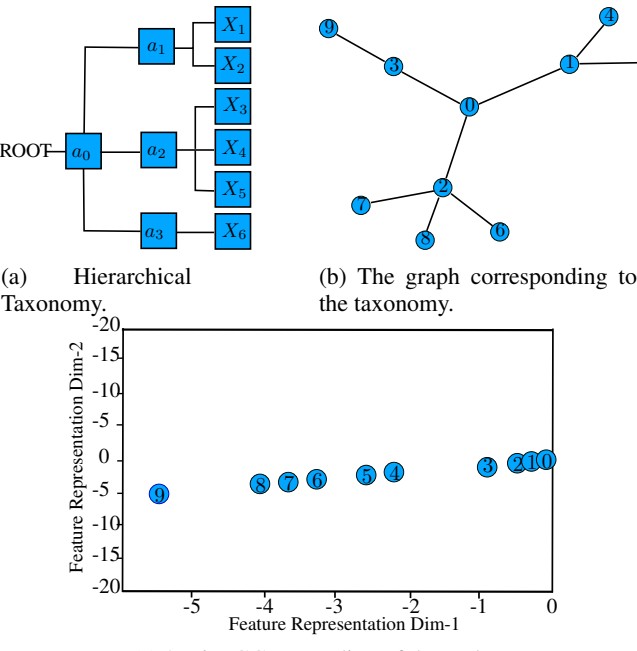

(a) Hierarchical Taxonomy.

(b) The graph corresponding to the taxonomy.

(c) 2-Dim GCN encoding of the nodes.

Figure 4: An illustration for GCN encoding of the nodes in the taxonomy with labeled nodes $a_0, .., a_3$, and unlabeled nodes $X_1, .., X_6$.

the taxonomy tree. Our method aims to mimic the Knowledge Distillation (Hinton, Vinyals, and Dean 2015) approach, where the hierarchical taxonomy serves as knowledge transferred from a teacher model (i.e., GCN model) to a student model (i.e., Symbolic-based model). In this method, we want the training algorithm (DNN) to not only rely on a supervised loss function but also consider prior domain knowledge encoded in a taxonomy tree. Therefore, a batch includes a few documents from the training data along with the taxonomy tree representing the hierarchical categories of the classification, which serves as the backbone of a larger graph $A$. In other words, $A$ is a graph generated by connecting the documents of a batch to the taxonomy tree. The workflow of the end-to-end training of BGCN is shown in Figure 5. The regularization term $\mathcal{L}_{reg}$, explicitly added to the existing loss function, is defined as:

$$\mathcal{L}_{reg} = \parallel P - H \parallel_2^2 \tag{5}$$

The generated graph $A$ is used to provide representations $H$ for batch of document in the same space of the predicted probabilities from a supervised DNN. Euclidean distance is measured as the regularization loss to be added to the training loss. The final loss function is:

$$\mathcal{L} = \mathcal{L}_0 + w \times \mathcal{L}_{reg} \tag{6}$$

where $\mathcal{L}_0$ is a Cross Entropy loss and $\mathcal{L}_{reg}$ is the regularization loss.

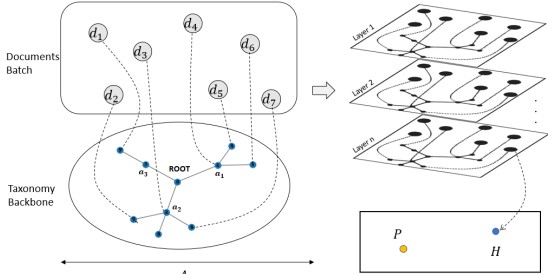

Figure 5: An illustration of BGCN training workflow on documents $D = \{d_1, d_2, .., d_7\}$ which are inter connected through a backbone graph, i.e., taxonomy.

## Experimental Results

### Taxonomy

To demonstrate the impact of taxonomy on both fine-grained and general classification, we adopt a policy. Our aim is to incorporate a wide range of categories at the top-1 level (excluding the root), while avoiding excessive depth in the taxonomy hierarchy. Specifically, we limit the taxonomy to three levels, thereby preventing the task from becoming solely focused on fine-grained classification.

### Data

In our experiments, we utilize datasets in two languages: Chinese ,i.e., In-house datasets, and English ,i.e., RCV1 and Amazon Product Review.

**In-house Data:** We utilize two imbalanced Chinese datasets from a private company. The first dataset comprises user query logs from a Shopping Mall, consisting of 84 classes. The taxonomy associated with this dataset has three levels: level 1 encompasses 18 domains, level 2 comprises 45 intents, and level 3 includes 84 sub-intents. The second dataset consists of user query logs from a Call Center Service, which consists of 134 classes. The hierarchical taxonomy for this dataset also has three levels: level 1 contains 5 domains, level 2 includes 24 intents, and level 3 (leaves/labels) contains 134 sub-intents.

**RCV1:** The Reuters Corpus Volume I (RCV1) is a widely recognized archive of over 800,000 manually categorized newswire stories provided by Reuters, Ltd. A new version of RCV1(RCV1-v2/LYRL2004) was provided by (Lewis et al. 2004), regarding this version of RCV1, we have generated a new dataset and a corresponding hierarchical taxonomy suitable for multi-class classification. We have created a hierarchical taxonomy with three levels tailored for multi-class classification. The first level comprises four categories, the second level consists of 33 categories, and the third level (leaves/labels) includes 53 sub-categories.

**Amazon Product Review:** This dataset consists of reviews from Amazon Since our focus is on multi-class flat classification, similar to the Reuters dataset, we filter out documents with multiple labels and create an adjusted hierarchical taxonomy accordingly. The updated taxonomy is

| Method | Accuracy% | Macro Avg F1% | Weighted Avg F1% |
|---|---|---|---|
| Baseline | 74.54 | 56.55 | 75.63 |
| +Symbolic-based | **76.23** | **60.84** | **77.39** |
| +GCN-based | 76.10 | 59.46 | 77.32 |

Table 1: A comparison of the methods in Supervised fashion on Call Center Service dataset.

| Method | Accuracy% | Macro Avg F1% | Weighted Avg F1% |
|---|---|---|---|
| Baseline | 93.80 | 81.69 | 93.57 |
| +Symbolic-based | **94.12** | 83.60 | **93.79** |
| +GCN-based | 93.99 | **86.11** | 93.74 |

Table 2: A comparison of the methods in Supervised fashion on Shopping Mall dataset.

organized into three levels: the first level includes 22 categories, the second level consists of 116 categories, and the third level (leaves/labels) encompasses 300 sub-categories.

## Baseline Method

**No Constraint:** LLMs have demonstrated significant improvements in machine learning performance, particularly in classification problems. It has yielded remarkable results even in few/zero-shot learning scenarios. To compare our results, we consider a LLM as the baseline without utilizing any constraints or taxonomy. It is a fine-tuned BERT model with two stacked layers on the *pooler_output*, employing *tanh* and *softmax* activation functions, respectively. We utilize two models from Hugging Face platform: *bert-base-cased* for English datasets, and *hfl/chinese-bert-wwm* for Chinese datasets.

**One-hot Constraint:** It is challenging to demonstrate the impact of explicit injected knowledge in pre-trained models, e.g., BERT, because of its already trained implicit knowledge. Therefore, to emphasize the effectiveness of integrating taxonomy knowledge into machine learning, we also employ the exactly-one or one-hot constraint, as presented in (Xu et al. 2018). This constraint captures the encoding used in multi-class classification, stating that for a set of indicators $X = \{X_1, ..., X_n\}$, one and exactly one of those indicators must be true, with the rest being false. The logical function for three variables can be expressed as $(x_1 \wedge \neg x_2 \wedge \neg x_3) \vee (\neg x_1 \wedge x_2 \wedge \neg x_3) \vee (\neg x_1 \wedge \neg x_2 \wedge x_3)$.

| Method | Accuracy% | Macro Avg F1% | Weighted Avg F1% |
|---|---|---|---|
| Baseline | 94.10 | 81.22 | 93.99 |
| +Symbolic-based | **95.16** | 82.45 | **95.08** |
| +GCN-based | 94.49 | **83.55** | 94.41 |

Table 3: A comparison of the methods in Supervised fashion on Reuters dataset.

## Ablation Study

We conduct an ablation study on our proposed approach. We refer to our base model as the Tax-based model and consider the following variant:

| Method | Accuracy% | Macro Avg F1% | Weighted Avg F1% |
|---|---|---|---|
| Baseline | 52.62 | 41.67 | 51.78 |
| +Symbolic-based | **53.77** | **42.14** | **52.64** |
| +GCN-based | 53.58 | 41.71 | 52.58 |

Table 4: A comparison of the methods in Supervised fashion on Amazon dataset.

| Method | Accuracy% | Macro Avg F1% | Weighted Avg F1% |
|---|---|---|---|
| Tax-based$_{Symbolic-based}$ | **+2.12** | **+3.66** | **+2.11** |
| Tax-based$_{GCN-based}$ | +0.59 | +2.72 | +0.71 |
| Tax-L1-based | +0.07 | +1.09 | +0.06 |

Table 5: The results of the ablation study.

| Portion | Method | Accuracy% | Macro Avg F1% | Weighted Avg F1% |
|---|---|---|---|---|
| 20% | Baseline | 65.94 | 43.58 | 64.95 |
| | +Symbolic-based | **70.51** | **52.85** | **70.12** |
| | +GCN-based | 68.00 | 50.28 | 67.74 |
| 30% | Baseline | 66.07 | 46.93 | 65.13 |
| | +Symbolic-based | 69.09 | 54.29 | 69.04 |
| | +GCN-based | **70.38** | **54.68** | **70.25** |
| 40% | Baseline | 68.58 | 51.67 | 68.20 |
| | +Symbolic-based | **71.93** | **58.54** | **71.88** |
| | +GCN-based | 69.99 | 55.17 | 69.59 |

Table 6: A comparison of the methods in Semi-Supervised fashion on Call Center Service dataset.

| Portion | Method | Accuracy% | Macro Avg F1% | Weighted Avg F1% |
|---|---|---|---|---|
| 20% | Baseline | 87.67 | 57.54 | 86.93 |
| | +Symbolic-based | **90.88** | 66.12 | **90.09** |
| | +GCN-based | 90.56 | **67.18** | 89.84 |
| 30% | Baseline | 91.53 | 71.87 | 90.48 |
| | +Symbolic-based | **92.22** | 76.23 | 91.75 |
| | +GCN-based | **92.22** | **81.26** | **91.93** |
| 40% | Baseline | 92.22 | 75.70 | 91.73 |
| | +Symbolic-based | **92.68** | 76.48 | **92.20** |
| | +GCN-based | 92.22 | **76.78** | 91.83 |

Table 7: A comparison of the methods in Semi-Supervised fashion on Shopping Mall dataset.

| Portion | Method | Accuracy% | Macro Avg F1% | Weighted Avg F1% |
|---|---|---|---|---|
| 20% | Baseline | 91.23 | 68.51 | 90.94 |
| | +Symbolic-based | 91.60 | 71.88 | 91.38 |
| | +GCN-based | **91.70** | **74.94** | **91.43** |
| 30% | Baseline | 92.74 | 75.38 | 92.56 |
| | +Symbolic-based | **93.43** | 76.54 | **93.40** |
| | +GCN-based | 92.54 | **77.08** | 92.45 |
| 40% | Baseline | 93.26 | 76.52 | 93.09 |
| | +Symbolic-based | **93.53** | 79.93 | **93.42** |
| | +GCN-based | 93.48 | **80.26** | 93.35 |

Table 8: A comparison of the methods in Semi-Supervised fashion on Reuters dataset.

| Portion | Method | Accuracy% | Macro Avg F1% | Weighted Avg F1% |
|---|---|---|---|---|
| 20% | Baseline | 40.93 | 19.99 | 36.94 |
| | +Symbolic-based | **44.00** | **27.94** | **41.83** |
| | +GCN-based | 43.76 | 25.34 | 40.90 |
| 30% | Baseline | 44.66 | 26.37 | 42.34 |
| | +Symbolic-based | 46.45 | **32.02** | **44.89** |
| | +GCN-based | **46.64** | 29.97 | 44.40 |
| 40% | Baseline | 48.09 | 32.13 | 45.43 |
| | +Symbolic-based | **49.14** | **36.47** | **47.87** |
| | +GCN-based | 48.80 | 33.39 | 47.01 |

Table 9: A comparison of the methods in Semi-Supervised fashion on Amazon dataset.

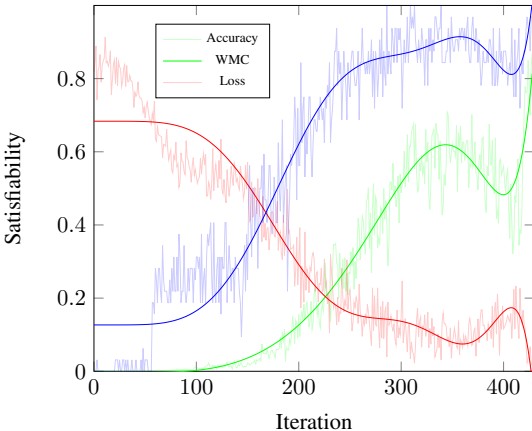

Figure 6: The satisfiability of WMC in semantic loss, training loss, and accuracy in first epoch.

**Tax-L1-based:** This variant of our base model includes only the first level $l_1$ of the taxonomy. Essentially, this variant is equivalent to the One-hot Constraint, as it represents the same formula proposed in (Xu et al. 2018). In this model variant, all upper levels of the taxonomy are removed, and only the leaf nodes are considered. To ensure a fair comparison with the One-hot Constraint, we evaluate the results only in a supervised fashion. To ensure reproducibility of our evaluation, we set all the *seeds* and run for one epoch. The average performance improvements, measured by three metrics across all four datasets, are presented in Table 5.

## Evaluation Measure

We use three measures: *Accuracy*, *Macro Average F1-score*, and *Weighted Average F1-score* to evaluate the obtained results. Specifically, for evaluating the effect of hierarchical taxonomy in imbalanced classification, we use *Macro Average F1-score*, which is the arithmetic mean of F1-scores per class. It does not use weights (i.e., number of true labels of each class) for aggregation of F1-scores per class, and this results in a bigger penalization when a model does not perform well with the minority classes. In all experiments, we obtain the results with/without injecting taxonomy into the existing data-driven loss function. We run all experiments for 10 epochs, with a batch size of 32. The experiments are repeated 3 times, and the best result for each method and baseline is selected.

In semi-supervised learning, we define a policy to generate datasets in which some of the target labels, i.e., leaves in taxonomy, for documents and some of internal nodes in the taxonomies, are randomly masked/removed. For example, 20% semi-supervised learning means that in the leaves of the taxonomy, which also correspond to the target labels, 80% of the data rely on the taxonomy. Moreover, from the 80% unlabeled data in this case, 40% of it relies on level 2 of the taxonomy, and the rest relies on first level. Respectively, in 30% and 40% semi-supervised learning, it is increased by 10% less relying on taxonomy in the leaves, and 20% on level 2. To examine the pure effect of taxonomy, we do not consider the taxonomy knowledge for both the labeled and unlabeled data in semi-supervised learning (Tables 6, 7, 8, and 9). We only take the knowledge from taxonomy into account for unlabeled data, and separately in different experiments (Tables 1, 2, 3, and 4), it is considered for all data.

Injecting the abstracted taxonomy into the existing loss function provides a learning signal on unlabeled samples by forcing the underlying learner to make decisions that satisfy the constraint. Figure 6 shows the satisfiability of the regularization term (i.e., WMC in semantic loss) of the training loss function in the first epoch, together with the training loss and the accuracy. The results indicate that as we reason on the hierarchical taxonomy, we will see improvement in categorizing documents.

Overall, the proposed methods consistently yield superior results in both supervised and semi-supervised scenarios when compared to the baseline. Notably, the substantial impact of the taxonomy becomes evident in the semi-supervised context, as demonstrated in tables (Tables 6, 7, 8, and 9), with a clear emphasis on improving the Macro Average F1-score. While *Symbolic-based* method demonstrates superior results, particularly in the accuracy of learners in supervised and semi-supervised scenarios, Tables 2, 3, 7, and 8 reveal that the *GCN-based* method outperforms in terms of Macro Average F1-score. This observation holds true even when considering that the size of the hierarchical taxonomies generated for the Shopping Mall and Reuters datasets is smaller than that of the other two datasets in our experiments, all while using a fixed batch size. These observations offer valuable insights from our experiments, highlighting two key takeaways. First, the *Symbolic-based* approach exhibits scalability for larger hierarchical taxonomies, showcasing its potential for handling more extensive taxonomic structures. Second, the *GCN-based* method proves to be particularly effective in addressing long-tailed problems, showcasing its utility in scenarios with imbalanced data distributions.

However, as shown in our experiments, the effect of the taxonomy on the model's performance reduces increasing of the labeled samples. As Table 5 shows, the one-hot constraint does not significantly affect performance, and it is almost zero because the constraint is always satisfied by existing data-driven loss, and it can perfectly fit training data. Despite this satisfaction, we can still see the effect of the taxonomy on overall performance, specially in minor classes, even in fully supervised learning Table 1 and 2. This is because of the hierarchical structure of the constraint, which alleviates model output distributions over the classes to allow conditioning on upper concepts.

## Conclusion

In this paper, we aimed to explore several key challenges in deep learning: reasoning, semi-supervised learning, and the long-tailed issue. We developed two methods to represent and integrate a hierarchical taxonomy of labels into the loss function of a flat classifier. We demonstrated the effect of these methods in supervised and semi-supervised learning. Moreover, our experimental evaluations show that integrating a well-designed hierarchical taxonomy into the learning

algorithm of a neural network effectively guides the learner to achieve significant results on long-tailed problems.

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
