# OpenReview forum: "TaxoKnow: Taxonomy as Prior Knowledge in the Loss Function of Multi-class Classification"
_AAAI.org/2024/Workshop/NuCLeaR — NuCLeaR 2024_

### Official Review · Reviewer_tWjk · 2023-12-08
**TaxoKnow: Taxonomy as Prior Knowledge in the Loss Function of Multi-class Classification**

**Rating:** 6
**Confidence:** 5

**Review:**

Scope:
The manuscript introduced a “Neuro Symbolic AI” and “Graph Neural Network” to solve the challenges of deep learning including reasoning, semi-supervised learning, and long-tailed issue. They proposed a integrated hierarchical taxonomy of class labels in loss function of classifier which help in fine grain classification of text documents.

Strength:
1.	The manuscript introduced the method to integrate a well-designed hierarchical taxonomy into the learning algorithm of a neural network for guiding the learner to achieve significant results in classification problem.

2.	New loss function was developed to trace loss in computing both semantic loss and its gradient.

3.	The authors have written a good quality of paper and covering all the relevant details to address the understanding of reader. The concepts were explained in good way and limitations of existing supervised classification and semi supervised methods were discussed in details.

4.	Detailed experimental results were reported in ablation study.

5.	The experimental results were reported on two standard English datasets and one self created Chinese dataset including, English RCV1 and Amazon Product Review and Chinese In-house datasets. Two languages were used to evaluate the performance of system.

Weakness:

1.	Architecture diagram of the proposed methodology is not clear. The author is suggested to redraw the architecture diagram highlighting the contribution of author in existing area of research.

2.	The result section is weak and rewrite the headings in results section to add more meaning. Author need to further elaborate their findings in better way. The ablation study need to be discussed and every table need to be explained.

3.	The related work section is  on weaker side and  author is suggested to update it with recent research paper to solve class imbalance problem as listed below:

Nida, N., Yousaf, M.H., Irtaza, A. and Velastin, S.A., 2022. Video augmentation technique for human action recognition using genetic algorithm. ETRI Journal, 44(2), pp.327-338.

Seo, K., Cho, H., Choi, D. and Park, J.D., 2022. Implicit Semantic Data Augmentation for Hand Pose Estimation. IEEE Access, 10, pp.84680-84688.

Škrlj, B., Martinc, M., Lavrač, N. and Pollak, S., 2021. autoBOT: evolving neuro-symbolic representations for explainable low resource text classification. Machine Learning, 110, pp.989-1028.

DeLong, L.N., Mir, R.F., Whyte, M., Ji, Z. and Fleuriot, J.D., 2023. Neurosymbolic ai for reasoning on graph structures: A survey. arXiv preprint arXiv:2302.07200.


Overall assessment:
 Marginally above acceptance threshold. Experimental result section need to be rewritten to add more clarity and cohesion.

---

### Official Review · Reviewer_FbcD · 2023-12-09
**Ideas presented with detailed empirical evaluation**

**Rating:** 7
**Confidence:** 4

**Review:**

**Summary:** This work proposes to utilize class hierarchy in different setups using symbol manipulation.

**Strengths:**
- Thorough experimental evaluation.
- Clear presentation of the ideas.

**Weaknesses:**
- In some tasks, the gaps are comparable with baselines. Confidence measurement should be presented.

---

### Decision · Program_Chairs · 2023-12-11

Accept